# Effect of Non-Nutritive Sweeteners on the Gut Microbiota

**DOI:** 10.3390/nu15081869

**Published:** 2023-04-13

**Authors:** Andrea Conz, Mario Salmona, Luisa Diomede

**Affiliations:** Department of Molecular Biochemistry and Pharmacology, Istituto di Ricerche Farmacologiche Mario Negri IRCCS, Via Mario Negri 2, 20156 Milano, Italy; andrea.conz@marionegri.it (A.C.); mario.salmona@marionegri.it (M.S.)

**Keywords:** gut microbiota, non-nutritive sweetener, aspartame, acesulfame, sucralose, saccharin

## Abstract

The human gut microbiota, a complex community of microorganisms living in the digestive tract, consists of more than 1500 species distributed in more than 50 different phyla, with 99% of bacteria coming from about 30–40 species. The colon alone, which contains the largest population of the diverse human microbiota, can harbor up to 100 trillion bacteria. The gut microbiota is essential in maintaining normal gut physiology and health. Therefore, its disruption in humans is often associated with various pathological conditions. Different factors can influence the composition and function of the gut microbiota, including host genetics, age, antibiotic treatments, environment, and diet. The diet has a marked effect, impacting the gut microbiota composition, beneficially or detrimentally, by altering some bacterial species and adjusting the metabolites produced in the gut environment. With the widespread use of non-nutritive sweeteners (NNS) in the diet, recent investigations have focused on their effect on the gut microbiota as a mediator of the potential impact generated by gastrointestinal-related disturbances, such as insulin resistance, obesity, and inflammation. We summarized the results from pre-clinical and clinical studies published over the last ten years that examined the single effects of the most consumed NNS: aspartame, acesulfame-K, sucralose, and saccharin. Pre-clinical studies have given conflicting results for various reasons, including the administration method and the differences in metabolism of the same NNS among the different animal species. A dysbiotic effect of NNS was observed in some human trials, but many other randomized controlled trials reported a lack of significant impacts on gut microbiota composition. These studies differed in the number of subjects involved, their dietary habits, and their lifestyle; all factors related to the baseline composition of gut microbiota and their response to NNS. The scientific community still has no unanimous consensus on the appropriate outcomes and biomarkers that can accurately define the effects of NNS on the gut microbiota.

## 1. The Gut Microbiota

The human gut microbiota is a complex ecosystem consisting of over 1500 species of bacteria distributed across more than 50 phyla [1]. About 30–40 species comprise over 99% of the total bacterial population, with up to 100 trillion bacteria residing in the large intestine alone [2,3]. The microbiota composition varies between individuals and can be categorized as either eubiosis, a healthy microbiota with a balance of beneficial bacterial species [4], or dysbiosis, a microbial imbalance that can lead to pathological outcomes. Microbial communities are unique to each host and can change rapidly due to various environmental factors, including diet [5].

The human gut microbiota comprises various categories of microbes, including bacteria, archaea, eukarya, viruses, and parasites [6]. The gut microenvironment favors the growth of bacteria from seven predominant divisions: Firmicutes, Bacteroidetes, Actinobacteria, Fusobacteria, Proteobacteria, Verrucomicrobia, and Cyanobacteria [7]. The Bacteroidetes and Firmicutes are the dominant bacterial populations in the gastrointestinal tract, accounting for over 90% of the total population [8] (Figure 1). The phylum Bacteroidetes consist of nearly 7000 different species of Gram-negative bacteria, primarily from the genera *Bacteroides*, *Alistipes*, *Parabacteroides*, and *Prevotella* (Figure 1). The Gram-positive bacterial species belonging to the phylum Firmicutes comprise over 200 genera, including *Clostridium, Eubacterium*, and *Ruminococcus*, which are predominant in the gut [9]. Other genera in this phylum, such as *Lactobacilli, Staphylococci*, and *Enterococci*, are in smaller numbers [8] (Figure 1). The Actinobacteria phylum is less abundant than Bacteroidetes and Firmicutes and mainly comprises the Bifidobacterium genus [10] (Figure 1). 

The ratio between the Firmicutes and Bacteroidetes (F/B) has been reported to influence the maintenance of gut homeostasis and the onset of various pathologies [11,12]. The F/B ratio commonly varies between 0.1 when the Firmicutes are 1/10 of the Bacteroidetes to 10 when the Firmicutes are ten times the Bacteroidetes [13,14]. Alterations in the abundance of specific Firmicutes or Bacteroidetes species can affect the F/B ratio. For instance, an increased Firmicutes population leads to a higher F/B ratio and is often associated with obesity [11,15]. Conversely, an increased Bacteroidetes count lowers the F/B ratio, which is linked to inflammatory bowel disease (IBD) [11,16]. The balance of the intestinal ecosystem is crucial for maintaining the physiological function of the human body, and various therapeutic approaches aim to achieve a balanced F/B ratio [11]. However, while the F/B ratio can indicate dysbiosis in the gut microbiota, contradictory results in the literature make it difficult to relate the F/B ratio to health and specifically consider it a hallmark of obesity or IBD [12,13].

Individuals vary in their abundance of bacterial phyla regardless of their enterotypes. Some bacteria are considered “core microbiota” because they are consistent between individuals, while others are more similar to a “flexible pool” that can adapt to the host. The “flexible pool” is acquired from ingested food, water, and environmental components. The genetic material exchange between the “core” and “flexible” pool can enable the host to adapt to different environments or dietary habits [2]. For example, certain genes were found in a marine bacterium living in red algae, and these same genes were also found in the gut bacterium of Japanese individuals due to their consumption of seaweed that is traditionally used to prepare sushi [17]. 

The gut microbiota is essential for the proper functioning of the host organism. It helps protect against pathogens by colonizing mucosal surfaces and producing antimicrobial substances. In addition, it plays a critical role in digestion, metabolism [18], and the immune system [19] and controls the growth and differentiation of epithelial cells [20]. The microbiota can also impact insulin resistance and secretion [21,22] and affect the host’s mental and neurological functions by influencing brain-gut communication [23].

The gut microbiota inhabits and reproduces on the surfaces of the intestines, forming a stable environment that helps prevent the infiltration of harmful microorganisms [24,25]. Furthermore, the metabolic functions of the gut microbiota encompass the breakdown of indigestible substances through anaerobic fermentation, generating short-chain fatty acids (SCFAs). These SCFAs participate in glucose [26] and lipid metabolism, regulate appetite and support the immune system [27]. They are also a crucial energy source for intestinal epithelial cells, reinforcing the mucosal barrier [28,29]. Additionally, SCFAs are receiving significant attention for their potential to promote human health since studies have shown that they may have anti-inflammatory and chemopreventive properties, making them a promising candidate for tumor suppression [27]. Given the gut microbiota’s crucial role in preserving a healthy gut function [30,31] and overall human health, its disruption is often associated with various diseases, including autoimmune diseases [32], inflammatory bowel diseases [33], cardiovascular disease [30,31], allergies [34], obesity and diabetes [35].

## 2. Effects of Diet on Gut Microbiota

Several factors can affect the composition and function of the gut microbiota, including host genetics, age [36], the mode of birth [37], antibiotic treatments [38], as well as environmental factors, including built and socioeconomic environments [18,39,40]. Diet is also essential in either a beneficial or harmful way by modifying bacterial species and adjusting the metabolites produced in the gut [41]. Diet is an essential factor that affects the development of the gut microbiota in newborn infants as it adapts to varying nutrient availability. During early infancy, the gut microbiota comprises genes that facilitate the breakdown of oligosaccharides in breast milk. Simultaneously, the introduction of solid foods leads to an increased abundance of genes related to the metabolism of polysaccharides and vitamins [42]. After infancy, diet becomes one of the main keys to organizing the gut microbiota’s structure, shape, and variety [38].

Different diets have a distinct impact on gut microbiota. Vegetarian diets may be related to changing microbiota species, resulting in the dominance of Firmicutes and Bacteroidetes [43]. Dietary fiber consumption is essential to maintain the integrity of the gut mucosal barrier function [44]. By contrast, a diet rich in protein and fats have been correlated with an abundance of Bacteroidetes, such as *Bacteroides*, *Bilophila*, and *Alistipes*, that are able to resist disruptive antibacterial compounds in the bile and the suppression of Firmicutes [45]. Increasing consumption of this diet may lower immunity and increase susceptibility to infection and metabolic diseases [46].

The excessive intake of simple sugars can proceed beyond the small intestine’s ability to absorb carbohydrates, leading to the provision of easily available substrates for bacterial growth in the colon and distal small intestine [47]. In addition, diets high in carbohydrates lead to a significant reduction in microbial variety, accompanied by an increase in saccharolytic bacteria in fecal matter, which is not typically present in the distal colon, regardless of the glycemic index of various high-carbohydrate fruits [48,49]. Some studies have indicated that a diet high in sucrose can change the function of *Lactobacillus plantarum* in the gastrointestinal system and lead to sucrose-induced dysbiosis, characterized by a surge in Clostridia and Bacilli and significant declines in *Lactobacillus* spp., *Sphingomonas*, and *Klebsiella* [50,51].

Research in recent years has emphasized the interplay between the microbiota and the host and how the composition of the human gut microbiota can potentially impact the onset of various diseases, including metabolic syndrome, obesity, type 2 diabetes, and type 1 diabetes [52]. These findings suggest that changes in the composition and function of the gut microbiota, including those caused by diet, directly affect human health and are important in the onset of several diseases. The consumption of unhealthy diets that are high in saturated fat and refined sugar and a lack of physical activity has been associated with dysbiosis of the gut microbiota, leading to inadequate glycemic control [53]. Any changes in the gut microbiota’s number, composition, or quality may affect the different physiological roles of microbes and cause gut microbiota dysbiosis [54].

## 3. Effects of Non-Nutritive Sweeteners on the Gut Microbiota

NNS are synthetic or natural sweeteners that are several hundred to several thousand times sweeter than sucrose [55] but without or with very few calories. Each NNS has specific metabolic characteristics, including a peculiar sweetness intensity and the persistence of the sweet taste.

The inclusion of NNS as food additives and sweeteners is a recent addition to the human diet [56,57]. The general consumption of NNS has significantly increased since saccharin was authorized as the first NNS in 1970 [58], and to date, the employment of NNS has been prevalent as it provides a useful strategy for lowering calorie consumption and sugar levels [59,60]. There are variations across nations when it comes to the kinds of NNS deemed safe for human consumption, provided that the Acceptable Daily Intake (ADI) is adhered to and there is no established connection with cancer or other health problems [61,62,63]. For instance, the European Food Safety Authority (EFSA) approved more types of NNS, such as cyclamate, compared to the U.S. Food and Drug Administration (FDA). However, recent research has suggested that the consumption of NNS can affect some health conditions, such as glucose intolerance and cardiovascular disease [59,64,65,66]. 

The gut microbiota has been implicated in the modification of xenobiotics (e.g., pharmaceutical drugs) [67,68], and there is evidence that it can similarly modify NNS as well [69]. Recently the effect of NNS on the gut microbiota has been investigated because of their possible impact on insulin resistance, obesity, and inflammation [57]. Several studies evaluating the effects of NNS on human physiology observed that the gut microbiota acts as a mediator of their potential effects [70,71,72], raising new questions regarding their safety profile and the need to clarify their interactions with the host and the microbiota.

While the health risks associated with excess sugar consumption are well known [16,73], our understanding is limited on whether or how NNS affects human physiology, whether they act directly on the host or indirectly through the modulation of the gut microbiota.

NNS are widely used due to their low caloric content and their ability to not affect post-meal appetite and energy responses. This is particularly beneficial for diabetic patients and overweight/obese individuals seeking sugar substitutes. The digestive system plays a crucial role in breaking down dietary components and facilitating nutrient absorption to meet the body’s nutritional requirements [74]. On the other hand, the gut microbiota consists of microorganisms that have access to dietary components before, during, and after digestion and absorption and are involved in their fermentation. They help to extract energy from nutritional components that the host cannot access and produce additional nutrients, such as vitamins, that can be utilized by the host [75].

The impact of this community on host growth, nutrition, and health is now better understood [75,76] as more instances of historical relationships between people and certain gut microorganisms have been discovered [77,78]. There is significant interest in understanding how food additives interact with the gut microbiota, and recent research has provided solid evidence for the influence of this bacterial population on the bioavailability and degradation of xenobiotic chemicals [79,80]. NNS are one of the most widely used food additives [81], and there has been a lot of interest in figuring out whether and how these substances change the gut microbiome [70,82,83]. This has made it even harder to fully understand the mechanisms beneath the correlation between NNS intake and dysbiosis [84]. 

Recent research has suggested that the gut microbiota, the complex community of microorganisms residing in the gastrointestinal tract, may play a role in the metabolism of NNS and their effects on host health. However, only a few studies have investigated the mechanisms by which gut microbiota can metabolize NNS, showing that they can be used as a carbon source by some strains of gut bacteria, leading to changes in their metabolic activity [70,72]. Understanding the effects of NNS on the gut microbiota, particularly on those most commonly used, such as aspartame, acesulfame-K, sucralose, and saccharin, is critical for assessing their impact on human health. Due to the variety of NNS’ chemical structures and metabolism, their peculiar activities in the gut microbiome need to be better understood, too (Figure 2). 

Aspartame is structurally simple and consists of a dipeptide methyl ester containing two amino acids that are found widely in fruits, vegetables, nuts, and dairy products, namely, L-aspartic acid and L-phenylalanine (Figure 2) [85]. 

Upon consumption, gastrointestinal peptidases and esterases almost completely break down aspartame, resulting in negligible amounts of the compound entering the bloodstream [85,86]. An intestinal esterase, such as chymotrypsin, removes the methyl group and releases the natural dipeptide aspartyl phenylalanine. The microvillus membrane lining the small intestine then metabolizes this dipeptide into its component amino acids, which enter the circulatory system [87]. Phenylalanine in the liver can be converted to tyrosine by phenylalanine hydrolases. However, if the body does not require excess phenylalanine, it is eliminated in the urine [88,89]. Once in systemic circulation, phenylalanine can be distributed throughout the body, including the brain, where it plays an important role in normal growth and development [85,88]. In the brain, tyrosine can further be converted into neurotransmitters such as dopamine, norepinephrine, and epinephrine. Thus, as such, aspartame in its whole form cannot interact directly with colonic microbiota.

Acesulfame-K is a hydrophilic, organic acid derivative (Figure 2) that is absorbed almost entirely in the small intestine as an intact molecule and is distributed by the blood to different tissues. Without undergoing any metabolism, more than 99% of ingested acesulfame-K is excreted in the urinary tract within the first 24 h, with less than 1% eliminated in the feces both in animals and humans [89,90]. The rapid absorption and urinary excretion it demonstrates cause a negligible acesulfame-K concentration that reaches the fecal or colonic bacteria [89,90]. Therefore, it is improbable that this NNS could directly affect the colonic microbiota [89,91].

Sucralose, a disaccharide composed of 1,6-dichloro-1,6-dideoxyfructose and 4-chloro-4-deoxygalactose (Figure 2), has a very low absorption level (less than 15%) and is practically not metabolized. Therefore, after intake, more than 85% of sucralose reaches the colon, which is unchanged in all species, including humans [89,89,92]. The small proportion absorbed is then eliminated in the urine and is mainly unchanged, though two glucuronides of sucralose were also detected in a small proportion (approximately 2%) [93]. Although more than 85% of the ingested sucralose contacts the colonic microbiota, 94%, and 99% can be recovered in the feces without any structural change, thus indicating little to no intestinal metabolism [89,91]. 

Saccharin is an acid that dissolves in water (Figure 2), and it is more easily absorbed by animals with a lower stomach pH, such as humans and rabbits, compared to animals with a higher stomach pH, such as rats [94,95]. In humans, between 85% and 95% of ingested saccharin is absorbed as an intact molecule since it does not undergo gastrointestinal metabolism. Once absorbed, it binds to plasma proteins, is distributed throughout the body, and is eliminated in urine through active tubular transport [89,94,96]. A small percentage of non-absorbed saccharin is excreted in the feces, indicating that high concentrations of this NNS could change the composition of the intestinal microbial population [91]. 

This review critically reassesses these aspects and examines the possible impact of NNS ingestion on the gut microbiota. We made an electronic search in the literature from 2013 to January 2023 on the following databases: PubMed MEDLINE^®^ and Embase^®^. The search was performed on 28 January 2023 with global geographical coverage and time limits on the ProQuest DIALOG™ search engine: the name of the compound of interest “acesulfame” AND/OR “aspartame” AND/OR “saccharin” AND/OR “sucralose” (and any other terms used to refer to them, such as non-nutritive sweeteners), the outcome measures (for example, dysbiotic effects on the gut microbiota) and lastly, the subjects on which these studies were conducted (animals and/or humans). We found 117 studies acceptable. The articles were then screened following the Preferred Reporting Items for Systematic Reviews and Meta-Analyses (PRISMA) diagram. Two expert reviewers independently selected papers according to the inclusion and exclusion criteria, as summarized in Table 1.

For each publication, we extracted the following information: Publication details, such as the authors, year, and journal;The characteristics of the participants, such as the number of participants recruited, the number of participants included in the study, sex, age range, and health status;The study design, including the design and blinding;Intervention characteristics, such as NNS doses, intervention duration, washout period (if planned), NNS format, other intervention types, and doses;Control characteristics, including the presence or absence of a control and control doses;Outcome measures, which included 16S rRNA gene sequencing and metabolomics to identify changes in the gut microbiota, fasting glucose levels, insulin tolerance test, body weight gain, gene expression of pro-inflammatory markers, and SCFA.

Twenty-seven of the 117 publications were excluded because they were duplicate records, and 11 were not written in English. Thus 79 studies were screened for eligibility. The selection process applied is described graphically in the PRISMA flowchart in Figure 3 [97]. 

Twenty-one studies were excluded as they were not about the effects of NNS, alongside 18 studies because they did not evaluate the outcome on gut microbiota changes after NNS administration, and 7 studies because they were not in animal models or humans. Among the 33 studies found eligible, 9 were excluded because they were reviews. Twenty-four studies met all the inclusion criteria and were included in this review (Figure 3). We summarize here the results from the pre-clinical and clinical studies published that examine the effects of the most consumed NNS: aspartame, acesulfame-K, sucralose, and saccharin.

### 3.1. Effect of Aspartame

The impact of aspartame on the composition of microbiota has been studied in two pre-clinical rat studies and four human studies (Table 2).

In 2014, a study examined the effects of chronic low-dose aspartame consumption on metabolic and microbial parameters in a diet-induced obese animal model. Male Sprague-Dawley rats were given a standard (12% kcal fat) or high-fat (60% kcal fat) diet along with aspartame in their drinking water at 5–7 mg/kg body weight (b.w.)/day for 8 weeks, or drinking water alone as a control [71] (Table 2). Aspartame reduced calorie intake, weight gain, and improved body composition when the rats were given a high-fat diet. However, aspartame increased fasting glucose levels and hindered insulin-stimulated glucose disposal in rats on both standard and high-fat diets, regardless of body composition. An analysis of the gut bacterial composition in fecal samples revealed that aspartame increased the total bacteria, Enterobacteriaceae, and *Clostridium leptum* levels [71]. Additionally, the high-fat diet led to a decrease in Bacteroidetes and an increase in Firmicutes. In contrast, aspartame administration alongside the high-fat diet mitigated an increase in Firmicutes without significantly affecting Bacteroidetes. The authors concluded that although aspartame influenced the gut microbial composition, further research was needed to understand the implications of these changes on the development of metabolic diseases [71].

A more recent study involving pregnant female Sprague Dawley rats fed a high-fat/sucrose (HFSD) diet, with or without 5–7 mg/kg b.w. Aspartame supplementation for 18 weeks observed increased body fat in offspring at weaning from dams fed the HFSD diet with aspartame compared to those fed HFSD alone, along with an altered glucose tolerance [98] (Table 2). Mothers had greater levels of *Akkermansia muciniphila* and Enterobacteriaceae than their infants. The aspartame group showed decreased levels of *Enterococcaceae, Enterococcus*, and *Parasutterella*, as well as elevated levels of *Clostridium cluster IV* in the cecal microbiota. Once the offspring’s feces were transplanted into germ-free mice, the gut microbiota of the latter was changed, which also affected adiposity and glucose tolerance. Additionally, both female and male offspring had increased Porphyromonadaceae concentrations in their guts [98].

Food elements play a significant role in determining the makeup of the gut microbiota. Different microbiological compositions might result from variations in the overall calorie intake and the type of food ingested [35,45,70]. It is uncertain whether the changes in the intestinal microbiota observed in this study were caused by the NNS alone and not by modifications triggered in the intestine by the HSFD diet itself because the intestinal microbiota could be altered by a wide range of nutritional interventions, such as the reduced consumption of fiber, proteins, fat, and carbohydrates. 

The first study on human subjects was a cross-sectional study published in 2015, investigating the effects of high-intensity sweetener consumption on the gut microbiota [99]. The experiment included 31 healthy individuals (65% of whom were female, with an overall body mass index (BMI) of 24.3) who were instructed to keep a food journal for four days in a row before collecting fecal samples on the fifth day. Seven individuals had ingested aspartame in the preceding four days, with a mean intake of 62.7 mg/day, whereas 24 participants had never used aspartame. Similar to this, 7 of the 24 participants who did not ingest acesulfame-K had daily intakes ranging from 1.7 mg to 33.2 mg on average. These people weren’t the same ones that ate aspartame. None of the 31 participants utilized dietary saccharin, although three of them consumed both aspartame and acesulfame-K. According to the Healthy Eating Index-2005, the BMI, calorie consumption, total carbohydrate intake, and diet quality of aspartame consumers and non-consumers were comparable [100]. There were no variations between the users of either sweetener and the non-consumers of either in terms of BMI or dietary traits [99]. Moreover, there were no changes in the overall count of bacteria between those who used aspartame or acesulfame-K and people who did not. Yet compared to people who did not take any sweetener, there was a decrease in the variety of microorganisms (from 24 to 7 phyla) [99] (Table 2).

Another study investigated the in vitro effect of NNS on fecal bacteria. Fecal samples from 13 healthy volunteers who had not directly consumed aspartame or any other NNS were fermented for 24 h in batch cultures in the presence of an aspartame-based sweetener, sucralose, or stevia product at concentrations corresponding to the acceptable daily intake (ADI) for an adult male weighing 75 kg. Only for a subset of eight participants, 16S rRNA amplicon sequencing of the human gut microbiome and the quantification of dominant bacterial groups were conducted. The analysis of human fecal samples fermented with an aspartame-based sweetener rich in maltodextrin indicated a significant increase in *Bifidobacterium* and *Blautia coccoides* growth and a lower *Bacteroides/Prevotella* ratio. It must be underlined that this study lacked appropriate controls since the authors did not evaluate the effects of the sweetener rich in maltodextrin and aspartame alone [79]. 

The Richardson Institute for Functional Foods and Nutraceuticals at the University of Manitoba in Winnipeg, Canada, conducted a randomized, double-blind crossover and controlled clinical experiment to examine the impact of sucralose or aspartame intake on the diversity of the gut microbiota [101]. A 12-week diet plan was followed by 17 healthy volunteers with a BMI of 20–25 within the age range of 18–45. For the first 4 weeks, there was a baseline period during which no artificial sweeteners were ingested by any of the subjects. Nine subjects received aspartame, and eight received sucralose throughout weeks 5 and 6. All subjects completed a washout phase between weeks 7 through 10, where no artificial sweeteners were used. Finally, during weeks 11 and 12, every subject ingested the new sweetener.

The study’s participants were instructed to abstain from using any additional NNS and were informed of the hidden sources of NNS in various foods, drinks, and pharmaceuticals. They were also told not to consume any probiotic-containing meals or supplements [101]. According to the mean adult body weight, the quantity of aspartame and sucralose that each participant should consume each day to meet 14% of the ADI for aspartame and 20% of the ADI for sucralose was calculated to be 0.425 g of aspartame and 0.136 g of sucralose each day, respectively. These doses were determined by the normal soft drink consumption habits of men and women in Canada [102] (Table 2). The authors considered the dosage levels reasonable and realistic because they reflected consumers’ daily intake of approximately three cans of diet soda [101].

Microbiota and SCFAs were examined in the feces that had been sampled both before and after treatments. The most prevalent bacterial taxa (family and genus) did not alter in their median relative proportions between treatments with NNS and those without. After ingesting the NNS, neither the microbiota community structure nor the fecal SCFAs changed (Table 2). Thus, aspartame did not cause healthy participants to change in the gut microbiota or SCFAs after 2 weeks of a realistic daily intake [101].

In 2022, contrasting results were obtained from a multi-arm randomized controlled trial assessing the effect of aspartame and other NNS on human metabolic health and the microbiota. In particular, Suez and colleagues enrolled 120 participants in this trial, which was divided into four groups (20 subjects/group) receiving aspartame, saccharin, sucralose, or stevia. Two control groups received glucose or no supplement [103]. All NNS were administered as commercially available sachets containing 5.76 g of glucose and 0.24 g of aspartame daily in a dosage of two sachets administered three times a day. The control group was given 5 g/day of glucose or nothing at all (Table 2). The experiment was broken down into three parts: after baseline assessments of metabolic, metabolomic, and microbiological parameters across 7 days, individuals were exposed to NNS for 14 days. Then, supplementation was stopped, and participants were monitored for a further 7 days. To determine how the medication affected the gut microbiota, samples of the microbiota from the oral cavity and feces were taken at predetermined intervals [103]. Aspartame and all the other NNS significantly altered the human intestinal microbiota and significantly affected microbiota function compared to the control groups [103]. However, which bacterial species increased and/or decreased was not specified. Aspartame also affected the oral microbiota, reducing the abundance of *Porphyromonas* and *Prevotella nanceiensis* (Table 2) [103].
nutrients-15-01869-t002_Table 2Table 2Studies evaluating the effect of aspartame on the gut microbiota.REFERENCESPECIESASPARTAME (APM) DOSE AND EXPOSUREOUTCOMESCLINICAL OUTCOMESPalmnäs et al., 2014 [71]Obese RatsStandard or high-fat diet ± 5–7 mg/kg b.w./day (in drinking water) for 8 weeks↑ total bacteria, Enterobacteriaceae and *Clostridium leptum*APM with a high-fat diet reduced caloric intake and rats gained less weight than with APM and a standard diet.APM with a high-fat diet elevated fasting glucose (*p* < 0.05) and impaired insulin-stimulated glucose disposal (*p* < 0.05) vs APM with the standard diet.Nettleton et al., 2020 [98]Pregnant Rats and offspringHigh fat/sucrose (HFS) diet ± 5–7 mg/kg b.w./day for 18 weeksIn cecal microbiota: ↓ concentration of Enterococcaceae, *Enterococcus*, *Parasutterella*, and ↑ *Clostridium cluster IV*↑ concentration of *Porphyromonadaceae* in males and females obese–aspartame offspringMaternal APM intake with a HFS diet increased body fat in offspring at weaning (*p* = 0.066) and body weight long-term (*p* ≤ 0.05).Maternal APM/ HFS impaired glucose tolerance in male offspring at age 8 weeks (*p* ≤ 0.05).Frankenfeld et al., 2015 [99]Human62.7 mg/day for 4 daysNo difference in bacterial abundance↓ in bacterial diversity from 24 to 7 phylaNo significant effectsGerasimidis et al., 2020 [104]In Vitro on human feces APM-based sweetener, 50 mg/kg b.w./day containing maltodextrin↑ *Bifidobacterium* and *Blautia coccoides* growth in feces↓ the *Bacteroides/Prevotella* ratio in fecesNot determinedAhmad et al., 2020 [101]Human 0.425 g/day for 2 weeksNo difference in microbiota community structure and fecal SCFAsSCFAs were not affected by APM and sucralose.Suez et al., 2022 [103]Human0.24 g/day APM + 5.76 g/day glucosecontrols: 5 g/day glucose or no supplementGut microbiota alterations↓ abundance of *Porphyromonas* and *Prevotella nanceiensis* in the oral microbiotaNo significant effects on glycemic response between the two groups (*p* = 0.076).

### 3.2. Effect of Acesulfame-K

The effects of acesulfame-K on the gut microbiota and fecal metabolic profiles were explored in animals (Table 3). In 8-week-old CD-1 mice, acesulfame-K was administered by gavage for 4 weeks, at 37.5 mg/kg b.w./day, and was investigated by 16S rRNA sequencing and gas chromatography-mass spectrometry metabolomics [105]. *Bacteroides* were greatly increased in acesulfame-K-treated male mice, with significant changes in two other genera, *Anaerostipes* and *Sutterella*. In female mice, acesulfame-K dramatically lowered the relative abundance of numerous genera, including *Lactobacillus*, *Clostridium*, an unassigned *Ruminococcaceae* genus, and an unassigned *Oxalobacteraceae* genus, and increased the abundance of *Mucispirillum* [105]. High gender-specific body weight gain, shifts in the gut bacterial community composition, an enrichment of functional bacterial genes related to energy metabolism, and fecal metabolomic changes were also observed. In particular, acesulfame-K increased body weight gain in male but not female mice [105]. 

Uebanso and colleagues obtained different results, reporting that the intake of 15 mg/kg b.w./day acesulfame-K, corresponding to the maximum ADI level, did not affect the relative amount of *Clostridium cluster* XIVa in the fecal microbiota [106]. In this study, 4-week-old male mice were allowed free access to a standard diet (AIN93G; Oriental Yeast) and were treated for 8 weeks with a solution containing 15 mg/kg b.w./day acesulfame-K or the same volume of distilled water [106]. Acesulfame-K did not increase food intake, body weight gain, liver weight, or fat in the epididymis or cecum [106] (Table 3).

Recent studies have indicated that 0.1 or 0.2 mg acesulfame-K (concentrations near the upper limit of human ADI), together with sucralose in pregnant mice, caused metabolic and microbiota alterations in the progeny [107] (Table 3). In particular, there was an increase in Firmicutes and depletion of *Akkermansia muciniphila*, which is a beneficial bacterium inversely correlated with fat mass gain, type 1 diabetes, and inflammatory bowel syndrome [107]. There was also an increase in the variety of species in the microbiota. 

In a study in 2021 in which C57BL/6J mice received 150 mg/kg b.w./day acesulfame-K for 8 weeks, acesulfame-K was reported to induce dysbiosis and intestinal injury with enhanced lymphocyte migration to the intestinal mucosa. Decreases in the levels of *Clostridiaceae*, *Lachnospiraceae*, and *Ruminococcaceae* in the gut microbiota were also observed (Table 3) [108]. 

These findings reported in the pre-clinical studies considered here suggest a dose-dependent relationship between acesulfame-K intake and the dysbiotic events in the gut microbiota of mice. 

In 2022, Murali and colleagues, to assess the influence of some NNS on fecal bacterial composition, treated Wistar rats of both sexes for 28 days with 40 mg/kg b.w./day and 120 mg/kg b.w./day acesulfame [109]. No mortality, abnormalities, or signs of clinical toxicity were observed in any of the animals in either dose group. The body weights of the animals and food consumption rates remained roughly the same throughout the 28-day study. No treatment-specific influence in either the dose groups of acesulfame-Kwas observed in the fecal 16S gene sequencing analysis in male and female rats compared to the respective controls. This is in line with the work of Uebanso et al., who demonstrated that acesulfame-Kat the maximum ADI of 15 mg/kg b.w. did not alter the gut microbiota. Murali et al. found that in males treated with acesulfame-K, fecal metabolites, predominantly amino acids, carbohydrates, lipids, fatty acids, and related classes, changed. However, when these metabolites were compared with key indicators of microbiota-associated changes in Wistar rats, based on different classes of antibiotics in 28-day studies [110,111], none were significantly altered in acesulfame-K-treated rats [109]. 

A cross-sectional study [99] indicated that acesulfame-K consumption did not cause notable differences in microbiota profiles and did not predict functional capacity across non-consumers and consumers [99]. In this study, as described in detail before, seven participants had consumed acesulfame-K in the four previous days with an average intake of 1.7 to 33.2 mg/day (Table 3). Acesulfame-K consumers were similar in BMI, energy intake, added sugar intake, and diet quality to non-consumers [99].

### 3.3. Effect of Sucralose

Given that the microbial metabolism of sucralose is practically null, for studies reporting an effect on the intestinal microbiota, it should be verified whether the experiments were conducted using pure sucralose or a commercial tabletop formulation since the latter usually contained about 1% sucralose and 99% of carriers were maltodextrins [91].

In this regard, Rodriguez-Palacios and colleagues gave Splenda^®^ (sucralose maltodextrin, 1:99, *w*/*w*) to SAMP1/YitFc (SAMP) mice to quantify the impact of a 6-week supplementation on the severity of Crohn’s disease-like ileitis and intestinal microbiota alterations [112]. Mice were given 3.5 mg/mL of Splenda^®^ in drinking water. SAMP and AKR mice (a strain widely used in cancer research for their high leukemia incidence and from which SAMP mice are derived) were treated with plain water for 6 weeks and were used as negative controls. Fecal bacteriome changes were subsequently assessed using a 16S rRNA microbiome.

The authors initially examined the gut metagenome profiles of AKR and the SAMP “experimental mouse” colony before investigating the effects of Splenda^®^. They collected and examined feces from six mice (three males and three females) aged 7, 22, and 50 weeks. By comparing SAMP mice to AKR mice, metagenomics showed a large rise in the Bacteroidetes phylum at the phylum level. At the class level, *Sphingobacteria* and *Bacteroidia* were more prevalent in SAMP mice than AKR mice within the family Bacteroidetes. Collectively, *Bacteroidia* was more often changed in the SAMP mice compared to *Sphingobacteria*, where only one genus out of six was elevated. The 16S microbiome research revealed that Splenda^®^ treatment resulted in a highly extensive promotion of bacterial species throughout the five classes of the Proteobacteria phylum, which was the most consistent impact (*Alphaproteobacteria*, *Betaproteobacteria*, *Epsilonproteobacteria*, *Deltaproteobacteria*, and Gammaproteobacteria). Moreover, Splenda^®^ stimulated the substantial development of *Escherichia coli* at the cost of streptococcus-like bacteria in the feces of mice [112].

In mice given 5 mg/kg b.w./day sucralose for 3 or 6 months, there was an enrichment of intestinal bacterial pro-inflammatory genes and a disruption in fecal metabolites, in addition to an increase in hepatic pro-inflammatory gene expression observed after 6 months of treatment (Table 4). These data suggest that sucralose at a dose level close to the human ADI may increase the risk of inflammation by disrupting the gut microbiota [72]. 

Dysbiosis from sucralose intake was also reported in another study by Wang and colleagues, where 5-week-old mice were fed a normal or high-fat chow diet for 8 weeks, with a drinking solution containing 2.5% (*w*/*v*) sucralose. Based on water consumption, sucralose intake was ~3.3 mg/kg b.w./day for mice fed a normal diet and ~1.5 mg/kg b.w./day for those fed the high-fat diet [113]. In mice fed the standard diet with sucralose, there was an increase in Firmicutes, which further increased when sucralose was provided in the high-fat diet (Table 4) [113]. 

However, the Uebanso study reported that, in mice, pure sucralose (14.2 mg/kg b.w./day) for 8 weeks reduced the relative amount of *Clostridium cluster XIVa* in the fecal microbiota [106]. A dysbiotic effect was also observed in the pups of dams treated for 6 weeks with sucralose together with 0.1 or 0.2 mg/kg b.w./day acesulfame-K; in these, the intestinal levels of Firmicutes doubled, including the *Clostridiales* families *Lachnospiraceae* and *Ruminococcaceae* (e.g., *Oscillospira*) [107] (Table 4).

According to these findings, Dai and colleagues examined the impact of maternal sucralose consumption on the propensity of kids to develop hepatic steatosis as adults [114]. A sucralose solution of 0.1 mg/mL was administered to C57BL/6 pregnant mice that were randomly assigned to either the maternal sucralose group or the maternal control group for the duration of gestation (3 weeks) and lactation (3 weeks). All young were fed a controlled diet after weaning for 8 weeks, followed by a high-fat diet for another 4 weeks. The 12th week was then used to evaluate the gut microbiota. After a 4-week high-fat diet, the authors found that maternal sucralose intake worsened intestinal dysbiosis in 12-week-old kids. Yet, Firmicutes and Proteobacteria increased in relative abundance. Bacteroidetes decreased concurrently in the maternal sucralose and control groups. Nevertheless, Proteobacteria’s abundance dramatically decreased in the maternal sucralose group compared to the control group after 9 weeks of a high-fat diet [114]. Since no 16S rRNA gene sequencing analysis was conducted, there was no clear information on the difference in the gut microbiota composition between the maternal sucralose and control groups.

In the 2022 study by Zheng and colleagues, mice were treated with 0.0003–0.3 mg/mL of sucralose (0.3 mg/mL of sucralose was equal to the ADI of 5 mg/kg b.w./day established by the FDA) [115]. The liver was removed from each mouse after 16 weeks, weighed, and the contents of the jejunum, ileum, cecum, and colon were collected. While sucralose affected the function of the intestinal barrier, there was no change in body weight. Sucralose dramatically altered the makeup of the gut microbiota, particularly at concentrations of 0.3 mg/mL or above, which increased the numbers of potential pathogens such as *Tenacibaculum*, *Ruegeria*, and *Staphylococcus* in the jejunum, ileum, and colon. At this dose, sucralose also increased *Allobaculum*, which was reported to be positively correlated with diabetes. A reduction in *Lachnoclostridium* and *Lachnospiraceae* was also found in the cecum compared with the controls given in water) [115].

In 2019, the composition of the intestinal microbiota in healthy subjects consuming sucralose was examined for the first time in a randomized, controlled, double-blind study. A group of 34 healthy male volunteers received 780 mg/day of sucralose in a capsule or a capsule containing a placebo for 7 days (Table 4). The dose of sucralose corresponded to 75% of the ADI [116]. No changes in glycemic control, insulin resistance, and intestinal microbiota at the phylum level were seen in subjects receiving sucralose (Table 4) [116,117,118].

The results were different in another study in which fermentation for 24 h of fecal samples from 13 healthy volunteers, together with 5 mg/kg sucralose, increased the growth of *Bifidobacterium* and *Blautia coccoides* and reduced the *Bacteroides/Prevotella* ratio (Table 4). The production of long SCFA valeric acid also increased, indicating that sucralose affected the synthesis of SCFA [79]. 

The randomized, double-blind crossover and controlled clinical trial by Ahmad and colleagues [101] on volunteers treated for 2 weeks with 0.136 g/day sucralose found no differences in the gut microbiota composition (family and genus) before and after the intake [101] (Table 4).
nutrients-15-01869-t004_Table 4Table 4Studies evaluating the effect of sucralose on the gut microbiota.REFERENCESPECIESDOSE OF SUCRALOSE AND EXPOSUREOUTCOMESCLINICAL OUTCOMESBian et al., 2017 [72]Male mice5 mg/kg b.w./day for 3 or 6 monthsat 3 months: ↑ *Ruminococcus*;↓ *Lachnospiraceae*, *Dehalobacteriaceae*, *Anaerostipes*, *Staphylococcus*, *Peptostreptococcaceae*, *Bacillus*at 6 months: ↑ *Akkermansia*, *Turicibacter*, *Roseburia*, *Clostridiaceae*, *Christensenellaceae*;↓ *Streptococcus*, *Lachnospiraceae*, *Dehalobacteriaceae*, *Erysipelotrichaceae*at 3 months: Not reportedat 6 months: Increase in genes related to bacterial pro-inflammatory mediators in sucralose-treated mice (*p* < 0.01).Hepatic increase in the gene expression of pro-inflammatory markers, (*p* < 0.05).Uebanso et al., 2017 [106]Male mice15 mg/kg b.w./day for 8 weeks*↓* relative amount of *Clostridium cluster XIVa* in the fecal microbiotaIncrease in hepatic cholesterol and cholic acid.Rodriguez-Palacios et al., 2018 [112]SAMP1/YitFc mice3.5 mg/mL of Splenda^®^ (sucralose maltodextrin, 1:99, *w*/*w*) in the drinking water for 6 weeks↑ 5 microbial classes within the Proteobacteria phylum (*Alphaproteobacteria*, *Betaproteobacteria*, *Epsilonproteobacteria*, *Deltaproteobacteria*, and *Gammaproteobacteria*).↑Escherichia coli in fecal microbiotaPossible exacerbation of myeloperoxidase and intestinal reactivity in mice with a pro-inflammatory predisposition but not in healthy animals.Wang et al., 2018 [113]5-week-old Mice~3.3 mg/kg b.w./day + standard diet~1.5 mg/kg b.w./day + high-fat diet↑ Firmicutes↑ Further increase of FirmicutesBody weight loss in mice fed a standard (*p* < 0.0001), but not a high-fat diet (*p* = 0.1250) in the absence of differences in food intake, calorie intake, or water intake.Stichelen et al., 2019 [107]Pregnant miceSucralose + 0.1 or 0.2 mg Acesulfame-K for 6 weeks↑ Firmicutes in the gut microbiota of offspring↓ *Akkermansia muciniphila* in the gut microbiota of offspringA total of 0.2 mg of sucralose and 0.5 mg of acesulfame-K lowered the pups’ weight (*p* < 0.0001) and fasting glucose (*p* < 0.05) vs the control group.Dai et al., 2020 [114]Pregnant miceSucralose solution of 0.1 mg/mL for 6 weeksRelative abundance of Firmicutes and Proteobacteria elevated and Bacteroidetes reduced in maternal sucralose and control groupsAfter 9 weeks of HFD the abundance of Proteobacteria decreased more significantly in the maternal sucralose group than in the control groupHigher expression of proinflammatory cytokines in maternal sucralose vs the maternal control group (*p* < 0.05).Exacerbation of high-fat diet-induced hepatic steatosis in 12-week-old offspring, and increases in hepatic IL-6 and tumor necrosis factor-α (TNF-α) (*p* < 0.05), and lipid metabolism genes (*p* < 0.01).Zhengi et al., 2022 [115]Mice0.0003–0.3 mg/mL of sucralose↑ *Tenacibaculum*, *Ruegeria*, *Staphylococcus* in jejunum, ileum and colon area.↑ *Allobaculum*,*↓ Lachnoclostridium* and *Lachnospiraceae* in the cecum of the 0.3 mg/mL group miceNo difference in body weight and liver weight between the control and treated mice.Damage to intestinal barrier and goblet cells in the treated vs. control group (*p* < 0.01), and distinct lymphocyte aggregation in ileum and colon.Thomson et al., 2019 [116]Humans780 mg/day for 7 daysNo effectsNo difference in body weight.Glycemic control and insulin resistance were not affected.Gerasimidis et al., 2019 [104]In Vitro on human feces5 mg/kg for 24 h↑ *Bifidobacterium* and *Blautia* coccoides growth↓ the *Bacteroides/Prevotella* ratio↑ SCFANot determinedAhmad et al., 2020 [101]Humans0.136 g/day for 2 weeksNo effectsSCFAs were not affected by aspartame and sucralose.Méndez-García et al., 2022 [119]Humans48 mg/day for 10 weeks↑ Blautia coccoidesVolunteers drinking sucralose for 10 weeks, but not the controls, had a larger area under the curve of glucose (AUCG) than at the beginning of the study (*p* = 0.02).Volunteers drinking water or sucralose for 10 weeks had similar AUCs of insulin (AUCIs) to those at the beginning of the study.Suez et al., 2022 [103]Humans102 mg/day sucralose + 5900 mg/day glucose for 2 weeksControls: 5000 mg/day glucose or no supplementGut microbiota alterationsChanges in six *Streptococcus* species in the oral microbiotaSucralose raised the glycemic response compared with glucose (*p* = 0.004) and no supplement control groups (*p* = 0.001).

More recently, two additional studies were published. The first was an open-label, randomized clinical trial with 47 healthy volunteers who did not habitually consume sucralose-containing products and agreed to avoid consuming any non-caloric sweeteners during the study [119]. These subjects were also instructed to follow a balanced diet of vegetables, fruits, grains, proteins, and dairy products, which were recommended according to their physical activity level. They were monitored weekly by phone interviews [119]. Each day for 10 weeks, participants used bottles containing 60 mL of sterile water or 48 mg of sucralose dissolved in 60 mL of sterile water (Table 4). This daily dose of sucralose resembled the daily consumption of four packets of commercial Splenda^®^, representing less than 15% of the ADI set by the FDA and less than 5% of the ADI established by the Joint Food and Agriculture Organization (FAO)/World Health Organization Expert Committee on Food Additives (JECFA) [120]. At the end of the trial, bacterial DNA was isolated from stool samples of the control and intervention groups. There were no differences in the level of *Actinobacteria* and *Bifidobacterium longum*, as well as Bacteroidetes, between the control and sucralose groups. Volunteers drinking sucralose presented a significant decrease at the end of the study in the relative abundance of Firmicutes, particularly *Lactobacillus acidophilus*, compared to their basal level. However, no differences were observed in the level of *Firmicutes* between the control and treated groups at the end of the study. In contrast, a small but significant increase in *Blautia coccoides* was detected [119]. 

The second recent study investigated the effect of a 2-week consumption of 0.102 g/day sucralose with 5.9 g/day glucose (Table 4) [103]. Control subjects received 5.9 g/day glucose or nothing. A significant alteration of the human intestinal microbiota was observed in volunteers receiving sucralose, although the bacterial species that were impaired were not specified. Modifications of the oral microbiota were also observed in six *Streptococcus* species [103]; in particular, concentrations of *Streptococcus salivarius* and oral taxon 064 decreased; meanwhile, levels of *Streptococcus pneumoniae*, DORA-10, HMSC073F11, and KR increased.

### 3.4. Effect of Saccharin

In 2014 a study was published investigating the effect of saccharin on the gut microbiota of mice or humans [70]. The authors examined the fecal microbiota of mice before and after 5 weeks of treatment with 5 mg/kg b.w. saccharin by sequencing their 16S ribosomal RNA gene. Water or water supplemented with glucose (concentration not disclosed) was given as a control. Mice drinking saccharin had a distinct microbiota composition clustered separately from their starting microbiota configurations. Both the control groups were the water group at week 5 and the glucose-supplemented water at week 11. Compared to the controls, the microbiota of saccharin-consuming mice displayed considerable dysbiosis, with an increase in the relative abundance of bacteria belonging to the *Bacteroides* genus and *Clostridiales* order and a reduction in *Lactobacillus reuteri* [70]. 

In the same studies, the effect of saccharin in humans was examined in seven healthy volunteers (five males and two females, aged 28–36) who consumed 5 mg/kg b.w./day saccharin for 6 days. This dose corresponds to the FDA’s maximal ADI for saccharin. During the experiment, volunteers did not ingest any other sources of saccharin or NNS. At the end of the intervention, four of the seven volunteers developed impaired glucose tolerance (responders), and three did not (non-responders) [70], suggesting that there was an individual glucose response to saccharin that might be mediated by differences in gut microbiota composition. As shown by 16S rRNA fecal analysis, the microbiota configurations of saccharin responders clustered differently from non-responders before and after NNS consumption. Microbiotas from non-responders presented small changes in composition during the study week. In contrast, compositional changes were pronounced in responders. There was a 20-fold relative increase in *Bacteroides fragilis* (order *Bacteroidales*) and *Weissella cibaria* (order *Lactobacillales*) and an approximately 10-fold decrease in *Candidatus arthromitus* (order *Clostridiales*) [70]. 

To study whether this saccharin-induced dysbiosis has a causal role in generating glucose intolerance, the stools from responders and non-responder volunteers before (day 1) or after (day 7) saccharin exposure were transferred into germ-free mice. The transfer of stools collected on day seven from responder volunteers receiving saccharin, but not that collected on day 1, induced significant glucose intolerance in recipient mice. In addition, the microbiota of these transplanted mice replicated some of the donor saccharin-induced dysbiosis. No glucose intolerance was observed in mice transplanted with stools collected on day 7 from non-responders [70].

In 2017, another study reported saccharin’s influence on gut microbiota composition [121]. Bian et al. treated C57BL/6J male mice with saccharin dissolved in drinking water at 0.3 mg/mL for 6 months. To investigate changes in gut microbiota, 16S rRNA gene sequencing was employed in fecal samples collected at different time points. Saccharin induced significant changes in the mouse gut microbiota, which was manifested by gut bacteria alterations [121]. At the baseline, the relative abundance of bacteria did not significantly differ between the treatment and control groups; however, a significant distinction was observed at 3 or 6 months or both. In particular, eleven genera were significantly changed after a 3- and 6-month treatment, indicating the effect of saccharin on disrupting the dynamics of gut microbiota development. Specifically, *Sporosarcina*, *Jeotgalicoccus*, *Akkermansia*, *Oscillospira*, and *Corynebacterium* were significantly higher after a 3-month consumption; *Corynebacterium*, *Roseburia*, and *Turicibacter* were increased after 6 months. *Anaerostipes* and *Ruminococcus* were significantly lower after a 3-month consumption; *Ruminococcus*, *Adlercreutzia*, and *Dorea* were lower after 6 months of consumption [121]. Additionally, levels of hepatic pro-inflammatory genes, such as inducible nitric oxide synthase and TNF-α, increased after 6 months of treatment [121]. These results indicate that saccharin might perturb the gut microbiota, consistent with Suez et al.’s report [70].

No effects on gut microbiota were reported in a study in 2019 [122] (Table 5) in which dogs were fed for 10 days with a diet containing 5% cellulose, 5% fiber, and a prebiotic blend, 0.02% of SUCRAM^®^ (an artificial sweetener consisting of 50% saccharin in addition to neohesperidin dihydrochalcone) and eugenol, or 5% fiber, a prebiotic blend plus 0.02% of saccharin and eugenol. The diets containing saccharin did not affect the proportions of bacterial phyla or fecal microbial composition [122].

Similar data were obtained in 2021 by Serrano and colleagues, whose double-blind, placebo-controlled, parallel-arm study explored the effects of pure saccharin on gut microbiota and glucose tolerance in 46 healthy subjects [123]. The participants were randomized into four treatment groups for 2 weeks of capsules containing 400 mg/day of sodium saccharin, 670 mg/day of lactisole, 400 mg/day of sodium saccharin, and 670 mg/day of lactisole or 1000 mg/day of a pulp filler/placebo. In parallel, the authors ran a 10-week study administering pure saccharin at a high dose (250 mg/kg bw/day) in the drinking water of chow-fed mice with genetic ablation of sweet taste receptors (T1R2-KO) and wild-type (WT) littermate controls. In humans and mice, pure saccharin supplementation did not alter microbial diversity or composition at any taxonomic level in humans and mice alike, and none of the interventions affected glucose or hormonal responses to an oral glucose tolerance test or glucose absorption in mice [123] (Table 5). These results indicate that short-term saccharin consumption at the maximum acceptable levels established by JECFA did not modify healthy humans’ gut microbiota and glucose tolerance [123]. However, more recently, a randomized controlled trial reported that the daily intake of 180 mg saccharin and 5820 mg glucose for 2 weeks, but not glucose alone, altered the gut microbiota and reduced the relative abundance of *Fusobacterium* in the oral microbiota [103].

In Murali and colleagues’ 28-day oral toxicity study [109], saccharin was administered to Wistar rats. The saccharin was prepared in a 0.5% carboxymethyl cellulose (CMC) suspension at doses of 20 mg/kg b.w./day and 100 mg/kg b.w./day and administered by gavage to five rats per treatment group per sex. The saccharin doses, which were 4 and 20 times the ADI of 5 mg/kg body weight, did not alter bacterial diversity compared to the controls. Serrano et al., 2021 made a similar observation when an even higher dose of saccharin (250 mg/kg b.w.) was supplemented and did not cause any gut microbiota changes in either mice or humans [123].

Consistent with Serrano’s finding, even though Murali and colleagues [109] observed an overall increase in intra-group variability in the NNS-treatment groups, there were very few significant alterations in the 16S bacterial compositions. For example, whether greater fluctuations in the bacterial family *Verrucomicrobiaceae* were treatment-related is uncertain, as this bacterial family was also widely variable in the controls [109]. Lastly, the saccharin 100 mg/kg bw/day treatment group showed a clear sex-dependent effect on changed fecal metabolites, but overall there was no marked impact on the fecal metabolomes in either sex [109].

Two studies investigated the veterinary use of saccharin to prevent enteric diseases in piglets [124]. Male and female suckling Landrace X Large White piglets aged 28 days were maintained for 2 weeks on three different isoenergetic diets consisting of (1) a commercial wheat- and soya-based swine basal diet (Target Feeds Limited) containing 42% (*w*/*w*) hydrolyzable carbohydrates, (2) a basal diet containing 5% (*w*/*w*) lactose, or (3) a basal diet supplemented with 0.015 % (*w*/*w*) SUCRAM^®^. Adding saccharin to the basal diet increased the cecal *Lactobacillus* populations, particularly *Lactobacillus OTU4228* [124] (Table 5). The addition of SUCRAM^®^ to the standard diet was also reported as reducing post-weaning enteric disorders, enhancing health, and reducing the mortality of early-weaned piglets [125] (Table 5). The pyrosequencing of pig cecal 16S rRNA gene amplicons identified 25 major families encompassing seven bacterial classes, with *Bacteroidia*, *Clostridia*, and *Bacilli* dominating the microbiota in pigs maintained on a standard diet, which was modified in animals receiving a diet containing SUCRAM^®^. The most notable change was a significant increase in the *Lactobacillaceae* population abundance, almost entirely due to a single phylotype, designated *Lactobacillus OTU4228*. It was concluded that the artificial sweetener, modifying the gut microbiota composition, could influence bacterial community dynamics. 

## 4. Conclusions

The effects of NNS on intestinal flora composition have been a research topic since the late 1980s [126,127,128,129]. Different effects of NNS on the metabolism of the gut microbiota have been described until now. Although a negligible amount of ingested NNS can reach the intestine, few studies here reported indicate that the gut microbiota can metabolize them by producing a variety of biological effects summarized in Figure 4. NNS can be used as a carbon source by some strains of gut bacteria, leading to changes in their metabolic activity and modulating the production of SCFAs [70,72]. These compounds, such as acetate, propionate, and butyrate, have an impact on glucose metabolism and host metabolism and exert an anti-inflammatory effect [130,131,132]. Increasing inflammation in the gut could contribute to various diseases such as IBD. The immune function of the host could also be modulated by the ability of NNS to reduce the abundance of some beneficial gut bacteria, such as *Bifidobacterium* and *Lactobacillus* [133,134], or to increase the abundance of pathogenic bacteria, such as *Clostridium difficile* and *E. coli*, which can cause infections and inflammation in the gut [135,136]. NNS can also alter the expression of genes involved in bacterial metabolism, altering the composition and function of the gut microbial community. They were also reported to be able to affect the release of gut hormones and neurotransmitters, influencing gut motility, nutrient absorption, and the composition of the gut microbiome, thus inducing alterations in glucose metabolism. Some studies suggested that NNS can induce gut dysbiosis and inflammation by increasing levels of bile acids [137,138].

However, as indicated by the data summarized in this paper, numerous conflicting results have been reported indicating that the topic still needs to be debated. Pre-clinical studies conducted mainly on rodents have focused on the number of intestinal total anaerobic and aerobic bacteria, bacterial diversity, and the F/B ratio and have investigated the effects of fecal transplantation and the maternal intake of NNS on offspring. These conflicting results can be explained by a wide range of reasons, including differences in the administration of NNS to animals, the metabolism of NNS in animal species and humans, and the lack of a clear definition of dysbiosis or eubiosis.

Although some human trials have observed a dysbiotic effect of NNS, many randomized controlled trials have reported a lack of significant effects on the gut microbiota composition after exposure to NNS. The studies reviewed in this paper differed in the number of subjects involved, their dietary habits, and their lifestyle, all factors relating to the baseline composition of gut microbiota and their response to NNS. Therefore, it is still hard to define precisely whether the use of NNS causes significant changes in the gut microbiota, as so much conflicting information has been published. On the whole, studies investigating the impact of NNS on the gut microbiota conclude that while the community may be altered in response to NNS, there are differences across studies complicating specific interpretations and direct comparisons while raising questions about a potential mechanism of action behind these responses. 

The discrepant results reported in this review also raise questions about the relevance of individual nutrients, such as NNS, to microbiota changes. Many lifestyle factors, especially negative health-related behaviors (i.e., smoking, alcohol addiction, unhealthy eating habits) and reduced access to medical and dental care, impact the microbiome [18,39,40]. Furthermore, both the environment we live in and the built environment, including structures built by humans such as homes, workplaces, schools, and vehicles, can influence the composition of the gut microbiome [139]. It is, therefore, clear that the ingestion of NNS is only one of the multiple factors that can have a significant impact on the composition and variety of the microbiota. Further studies are needed to establish whether the consumption of NNS alone, at doses permitted by regulatory agencies, is the most relevant factor.

## 5. Future Directions

New efforts are needed to conduct pre-clinical and clinical well-designed studies that are aimed at establishing the potential dysbiotic effects of NNS. In particular, long-term, double-blind, placebo-controlled, randomized trials with appropriate doses and adequate subject sizes are required to assess the impact of NNS on intestinal microbiota and how they might affect major outcomes related to chronic diseases. Despite these previously unappreciated impacts of NNS, their value must be considered in relation to their role in limiting caloric intake as alternatives to sugar, supporting oral hygiene, and reducing risk factors for the development of caries. The value of NNS to efforts limiting the global health burden of obesity and obesity-related disease may well outweigh their potential effects on the human gut microbiota.

The impact of NNS on the intestinal microbiota is made even more complex by recent data suggesting that, thanks to their effects on bacterial cell membranes and cellular permeability, they could contribute to the spread of antibiotic resistance [140,141]. Aspartame, acesulfame-K, sucralose, and saccharin, at concentrations corresponding to the ADI established by the FDA for an individual with a body weight of 60 kg and lower than the threshold concentrations regulated by the Codex General Standard for Food Additives, resulted in them being able to promote the intra-and inter-genus spread between bacteria of antibiotic resistance genes in a dose-dependent manner [140,141]. In addition, aspartame, acesulfame-K, and sucralose, but not saccharin, raised mRNA expression levels of genes that are essential for replicating the resistant genes and their transfer from the donor to the recipient bacteria [140,141]. These findings suggest that the four most commonly used sweeteners might exert a potential antibiotic-like effect contributing to the spread of antibiotic resistance. However, it is important to underline that these data were obtained in vitro, in simplified experimental conditions, without reproducing the complex physiological situation occurring in vivo, and their relevance should be demonstrated by additional experiments.

## Figures and Tables

**Figure 1 nutrients-15-01869-f001:**
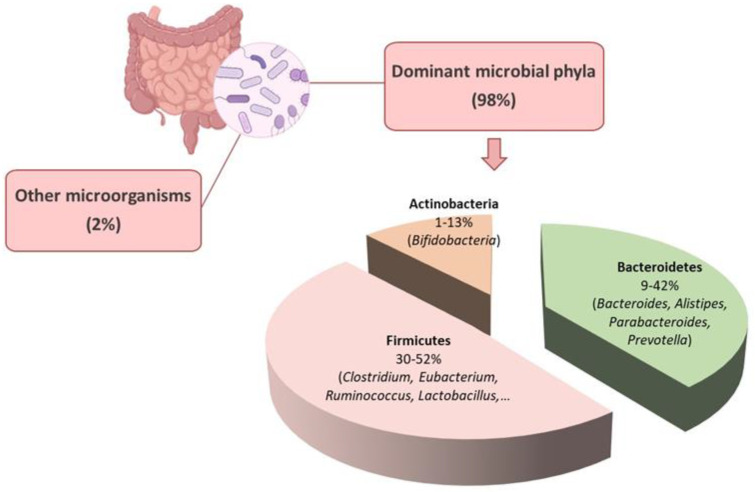
Phylum composition of gut microbiota. The dominant bacteria phyla in the gastrointestinal tract are Bacteroidetes, Firmicutes, and Actinobacteria, which account for about 98% of the total population. More than 90% of the microbial phyla are Bacteroidetes and Firmicutes. Created with BioRender.com.

**Figure 2 nutrients-15-01869-f002:**
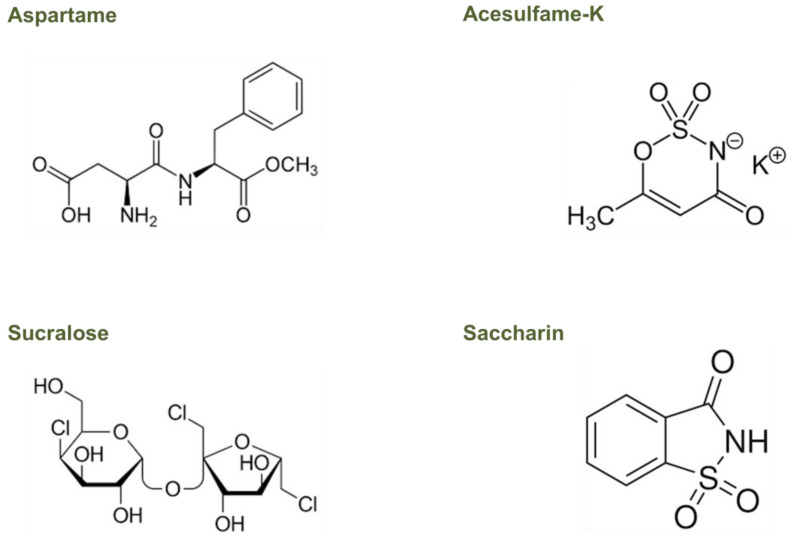
Chemical structures of NNS considered in this study.

**Figure 3 nutrients-15-01869-f003:**
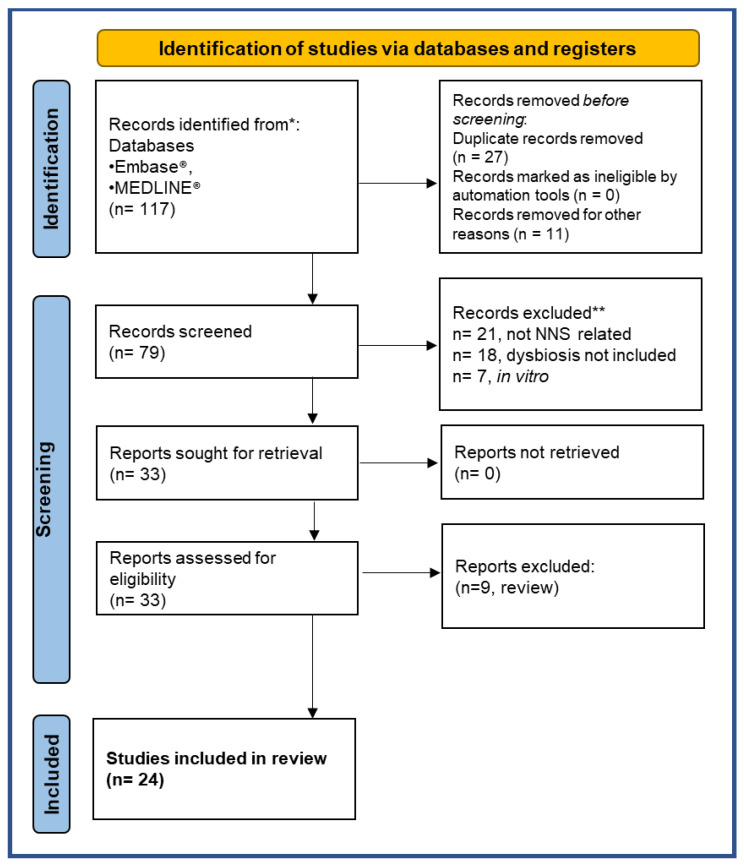
PRISMA 2020 flow diagram for new systematic reviews including searches of databases and registers only. * The number of records identified from each database or register searched (rather than the total number across all databases/registers). ** Records excluded by the researchers.

**Figure 4 nutrients-15-01869-f004:**
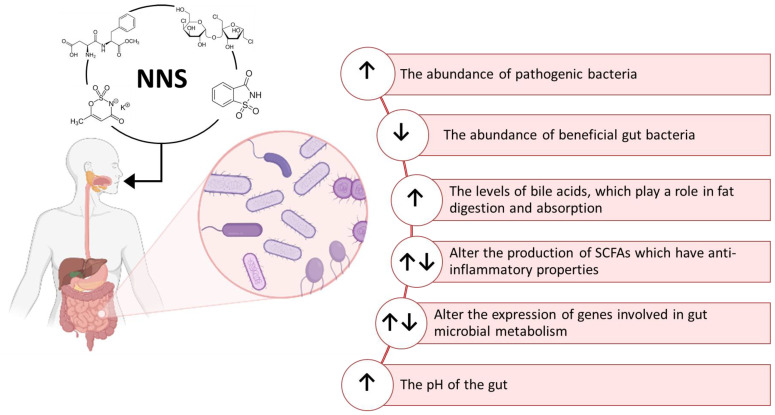
Overview of some of the documented effects of NNS in both humans and the gut microbiota metabolism. Created with BioRender.com.

**Table 1 nutrients-15-01869-t001:** Summary of inclusion and exclusion criteria employed for the studies’ selection.

	INCLUSION CRITERIA	EXCLUSION CRITERIA
**Intervention**	Oral intake of the NNS (aspartame, acesulfame-K, sucralose, saccharin)	Non oral NNS administration routes
**Comparator**	Placebo or standard diet	No control
**Outcome**	Expressed employing 16S RNA microbiota profiling or qPCR analysis	Not shown
**Study Design**	Randomized control trial, intervention trial (non-randomized, quasi-experimental), observational trial, pre-clinical in vivo study	Case reports, commentary, meta-analysis, systematic review, letters to the editor, in vitro or ex-vivo studies
**Publication**	Peer-reviewed paper	Congress abstracts, thesis reports, grey literature

**Table 3 nutrients-15-01869-t003:** Studies evaluating the effect of acesulfame-K on the gut microbiota.

REFERENCE	SPECIES	ACESULFAME-K DOSE AND EXPOSURE	OUTCOMES	CLINICAL OUTCOMES
Bian et al., 2017 [105]	Mice	37.5 mg/kg b.w./day for 4 weeks	In males ↑ *Bacteroides* significant changes in *Anaerostipes* and *Sutterella* In females: ↓ *Lactobacillus*, *Clostridium* ↑ *Mucispirillum*	Treated male mice had higher body weight gain than controls (10.28 g vs 5.44 g, *p* < 0.01). No difference between control and treated female mice.
Uebanso et al., 2017 [106]	Male mice	15 mg/kg b.w./day for 8 weeks	No effect on fecal microbiota	Acesulfame-K did not increase food intake, body weight gain, or liver weight.
Stichelen et al., 2019 [107]	Pregnant mice	0.1 or 0.2 mg acesulfame-K + sucralose for 6 weeks	↑ Firmicutes in the gut microbiota of offspring ↓ *Akkermansia muciniphila* in the gut microbiota of offspring	A total of 0.2 mg of sucralose and 0.5 mg of acesulfame-K lowered the pups’ weight (*p* < 0.0001) and fasting glucose levels (*p* < 0.05) vs the control group.
Hanawa et al., 2021 [108]	Male mice	150 mg/kg b.w./day for 8 weeks	Dysbiosis, intestinal injury with enhanced lymphocyte migration to the intestinal mucosa ↓ *Clostridiaceae*, *Lachnospiraceae*, and *Ruminococcacea*	Increased the expression of proinflammatory cytokines (*p* < 0.05). Lowered the expression suppressors of proinflammatory cytokines (*p* < 0.01).
Murali et al., 2022 [109]	Mice	40 or 120 mg/kg b.w./day for 4 weeks	No significant effect on the fecal microbiota	No mortality, no abnormalities, and no signs of clinical toxicity. Normal fecal consistency
Frankenfeld et al., 2015 [99]	Humans	1.7 to 33.2 mg/day for 4 days	No difference in bacterial abundance profiles	No significant effects

**Table 5 nutrients-15-01869-t005:** Studies on the effect of saccharin on the gut microbiota.

REFERENCE	SPECIES	DOSE OF SACCHARIN AND EXPOSURE	OUTCOMES	CLINICAL OUTCOMES
Suez et al., 2014 [70]	Mice	High fat diet + 5 mg/kg b.w./day of saccharin for 5 weeks	↑ *Bacteroides*, *Clostridiales*; ↓ *Lactobacillus reuteri*	Impaired glucose tolerance (*p* < 0.0002). Increased fecal levels of the SCFAs propionate (*p* < 0.1).
Bian et al., 2017 [121]	Male mice	0.3 mg/mL of saccharin in drinking water for 6 months	at 3 months: ↑ *Sporosarcina*, *Jeotgalicoccus*, *Akkermansia*, *Oscillospira*, and *Corynebacterium*; ↓ *Anaerostipes* and *Ruminococcus* at 6 months: ↑ *Corynebacterium*, *Roseburia*, and *Turicibacter*; ↓ *Ruminococcus*, *Adlercreutzia*, and *Dorea*	Increase in hepatic pro-inflammatory genes, inducible nitric oxide synthase, and TNF-α, (*p* < 0.05) after 6 months of treatment.
Nogueira et al., 2019 [122]	Dogs	0.02% of saccharin + eugenol for 10 days 0.02% of saccharin + eugenol + 5% fiber +prebiotic blend for 10 days 5% fiber + prebiotic blend diet for 10 days 5% cellulose for 10 days	No effects	No significant effects
Serrano et al., 2021 [123]	Mice (WT and T1R2-KO)	250 mg/kg bw/day	No significant effects	T1R2-KO mice were protected from age-dependent increases in fecal SCFA and the development of glucose intolerance vs WT mice.
Murali et al., 2022 [109]	Mice	20 or 100 mg/kg b.w./day for 4 weeks	No significant effect on the fecal microbiota	No mortality, no abnormalities, no signs of clinical toxicity Normal fecal consistency
Daly et al., 2014 [124]	Piglets	Basal diet for 2 weeks Basal diet + 5% (*w*/*w*) lactose for 2 weeks Basal diet + 0.015 % (*w*/*w*) saccharin for 2 weeks	↑ cecal *Lactobacillus* populations, particularly *Lactobacillus* OTU4228	No significant changes
Daly et al., 2016 [125]	Piglets	Basal diet for 2 weeks Basal diet + 0.015 % (*w*/*w*) saccharin for 2 weeks	↑ *Lactobacillaceae* population, particularly *Lactobacillus OTU4228*	No significant changes
Suez et al., 2014 [70]	Human	5 mg/kg b.w./day for 1 week	↑ *Bacteroides fragilis* and *Weissella cibaria* ↓ *Candidatus Arthromitus*	Four out of seven subjects developed significantly poorer glycemic responses 5–7 days after saccharin consumption, compared to their glycemic responses on days 1–4 (*p* < 0.001).
Serrano et al., 2021 [123]	Human	400 mg/day for 2 weeks 400 mg/day + 670 mg/day lactisole for 2 weeks 670 mg/day lactisole placebo	No significant effects	No significant effects
Suez et al., 2022 [103]	Human	180 mg/day saccharin + 5820 mg/day glucose for 2 weeks 5000 mg/day glucose for 2 weeks No supplement control	Gut microbiota alterations (not specified) ↓ relative abundance of *Fusobacterium* in the oral microbiota	Saccharin raised a glycemic response compared to glucose (*p* = 0.042) and no supplement control groups (*p* = 0.018).

## Data Availability

This is a review of the data available in the literature. No data were analyzed or generated during the study.

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
