# Peer review of "Effect of Non-Nutritive Sweeteners on the Gut Microbiota"

_nutrients, 2023, doi:10.3390/nu15081869_

Round 1
Reviewer 1 Report
This is a well written article that adds important insights to the scientific literature. The inclusion of multiple data extraction tables summarizing findings for different NNS was very helpful. The figures are also well done.
My main concern is to ensure that the statements regarding the impact of diet as the primary factor influencing the composition of the microbiome is accurate. Given the focus of this article, that conclusion precipitates; however, looking more broadly, I am wondering if the impact of dietary intake remains the primary factor. The following article offers some new insights into the other factors that mold the microbiome.
Gacesa R, Kurilshikov A, Vich Vila A, Sinha T, Klaassen MAY, Bolte LA, Andreu-Sánchez S, Chen L, Collij V, Hu S, Dekens JAM, Lenters VC, Björk JR, Swarte JC, Swertz MA, Jansen BH, Gelderloos-Arends J, Jankipersadsing S, Hofker M, Vermeulen RCH, Sanna S, Harmsen HJM, Wijmenga C, Fu J, Zhernakova A, Weersma RK. Environmental factors shaping the gut microbiome in a Dutch population. Nature. 2022 Apr;604(7907):732-739. doi: 10.1038/s41586-022-04567-7. Epub 2022 Apr 13.
Line 13 and 115: Add environment?
Line 123: Add source. Is this an accurate statement? Is it diet in combination of other factors?
Line 126: Add source.
Line 225: Suggest adding paragraph indent.
Line 227: Add source.
Author Response
Reviewer #1
Q1. My main concern is to ensure that the statements regarding the impact of diet as the primary factor influencing the composition of the microbiome is accurate. Given the focus of this article, that conclusion precipitates; however, looking more broadly, I am wondering if the impact of dietary intake remains the primary factor. The following article offers some new insights into the other factors that mold the microbiome: Gacesa R, Kurilshikov A, Vich Vila A, Sinha T, Klaassen MAY, Bolte LA, Andreu-Sánchez S, Chen L, Collij V, Hu S, Dekens JAM, Lenters VC, Björk JR, Swarte JC, Swertz MA, Jansen BH, Gelderloos-Arends J, Jankipersadsing S, Hofker M, Vermeulen RCH, Sanna S, Harmsen HJM, Wijmenga C, Fu J, Zhernakova A, Weersma RK. Environmental factors shaping the gut microbiome in a Dutch population. Nature. 2022 Apr;604(7907):732-739. doi: 10.1038/s41586-022-04567-7. Epub 2022 Apr 13.
A1. To address the reviewer’s comment, in addition to the diet, the impact of other factors on gut microbiota has been considered, also taking into account the suggested paper of Gacesa et al.
In the Section entitled “Effects of diet on gut microbiota”, on page 4, lines 108-110, the role of the environment on the gut microbiota has been mentioned by modifying the sentence as follows: “Several factors can affect the composition and function of the gut microbiota, including host genetics, age [36], mode of birth [37], antibiotic treatments [38] as well as environmental factors including built and socioeconomic environments [18,39,40].”
In the Conclusions, line 748, we have also added the following paragraph: “The discrepant results reported in this review also raise questions about the relevance of individual nutrients, such as NNS, to microbiota changes. Many lifestyle factors, especially negative health-related behaviors (i.e., smoking, alcohol addiction, unhealthy eating habits) and reduced access to medical and dental care, impact the microbiome [18,39,40]. Furthermore, both the environment we live in and the built environment, including structures built by humans such as homes, workplaces, schools, and vehicles, can influence the composition of the gut microbiome [139]. It is therefore clear that ingestion of NNS is only one of the multiple factors that can have a significant impact on the composition and variety of the microbiota. Further studies are needed to establish whether the consumption of NNS alone, at doses permitted by regulatory agencies, is the most relevant factor.”
The paper of Gacesa et al. has been cited as reference number 40.
Q2. Line 13 and 115: Add environment?
A2. In the Abstract, line 13, the word “environment” has been added. In addition, in the Section entitled “Effects of diet on gut microbiota”, on page 4, lines 108-110, the role of the environment on the gut microbiota has been considered by modifying the sentence as follows: “Several factors can affect the composition and function of the gut microbiota, including host genetics, age [36], mode of birth [37], antibiotic treatments [38] as well as environmental factors including built and socioeconomic environments [18,39,40].”
Q3. Line 123: Add source. Is this an accurate statement? Is it diet in combination of other factors?
A3. To make the sentence on page 4, line 117, more accurate specifying that the diet is not the only player involved in the composition of the microbiota, we have modified it as follows: “After infancy, diet becomes one of the main keys to organizing the gut microbiota's structure, shape, and variety [38].” The source has been also added.
Q4. Line 126: Add source.
A4. In the Section entitled “Effects of diet on gut microbiota”, on page 4, line 122, the following reference has been added: Tomova et al., Front Nutr 2019.
Q5. Line 225: Suggest adding paragraph indent.
A5. On page 8, line 277, the paragraph indent has been added, as requested.
Q6. Line 227: Add source.
A6. On page 8, line 280, the following reference has been added as number 97: Page et al. BMJ 2021.
Reviewer 2 Report
The manuscript by Andrea Conz et al. summarized the recent advances on the effect of non-nutritive sweeteners on the gut microbiota. The study is of interest to the readers and I have the following suggestions and comments:
1, The authors should add a new figure to summarize the effect non-nutritive sweeteners on the metabolism of the gut microbiota. For example, the production of SCFAs or bile acids. This would help to improve the manuscript.
2, The authors should add a "Future directions" section to discuss the future development in this field. More human studies are urgently needed. And this should be discussed and highlighted.
3, The authors should also give more thoughts about the mechanisms by which non-nutritive sweeteners changes the gut microbiota. Are the non-nutritive sweeteners being metabolized or consumed by the microbiota? How and Why? This should be discussed and this is very important.
4, The authors should add a new figure to show the chemical structures of the non-nutritive sweeteners. This would help the readers to understand the sweeteners more easily.
Author Response
Reviewer # 2
Q1. The authors should add a new figure to summarize the effect of non-nutritive sweeteners on the metabolism of the gut microbiota. For example, the production of SCFAs or bile acids.
A1. A new Figure 5, has been added, as requested and in the Conclusions, line 706, the following paragraph has been inserted to summarize the effects of NNS on both the human and the gut microbiota metabolism: “Different effects of NNS on the metabolism of the gut microbiota have been described till now. Although a negligible amount of ingested NNS can reach the intestine few studies here reported indicated that the gut microbiota can metabolize them producing a variety of biological effects summarized in Figure 5. NNS can be used as a carbon source by some strains of gut bacteria, leading to changes in their metabolic activity and modulating the production of SCFAs [70,72]. These compounds, such as acetate, propionate, and butyrate, have an impact on glucose metabolism, and host metabolism and exert an anti-inflammatory effect [130–132]. Increase inflammation in the gut, could contribute to various diseases such as IBD. The immune function of the host could also be modulated by the ability of NNS to reduce the abundance of some beneficial gut bacteria, such as Bifidobacterium and Lactobacillus [133,134], or to increase the abundance of pathogenic bacteria, such as Clostridium difficile and E. coli, which can cause infections and inflammation in the gut [135,136]. NNS can also alter the expression of genes involved in bacterial metabolism, altering the composition and function of the gut microbial community. They were also reported able to affect the release of gut hormones and neurotransmitters, influencing gut motility, nutrient absorption, and the composition of the gut microbiome thus inducing alterations in glucose metabolism. Some studies suggested that NNS can induce gut dysbiosis and inflammation by increasing the levels of bile acids [137,138].”
Q2. The authors should add a "Future directions" section to discuss the future development in this field. More human studies are urgently needed. And this should be discussed and highlighted.
A2. A new section entitled “Future Directions” has been added at line 757 to discuss the need for pre-clinical and clinical well-designed studies aimed at establishing the potential dysbiotic effects of NNS.
Q3. The authors should also give more thoughts about the mechanisms by which non-nutritive sweeteners changes the gut microbiota. Are the non-nutritive sweeteners being metabolized or consumed by the microbiota? How and Why? This should be discussed and this is very important.
A3. The mechanisms underlying the metabolism of NNS by microbiota remain still to be fully clarified. We found only a few studies on this topic that have been now better discussed in the manuscript (Suez et al. Nature 2014; Bian et al., Front Physiol 2017). The authors proposed that NNS can be used as a carbon source by some strains of gut bacteria, leading to changes in their metabolic activity and modulating the production of compounds with anti-inflammatory effects.
This point has been addressed, in the Section entitled “Effects of non-nutritive sweeteners on the gut microbiota”, on page 5, line 193, where the following paragraph has been added: “Recent research has suggested that the gut microbiota, the complex community of microorganisms residing in the gastrointestinal tract, may play a role in the metabolism of NNS and their effects on host health. However, only a few studies have investigated the mechanisms by which gut microbiota can metabolize NNS showing that they can be used as a carbon source by some strains of gut bacteria, leading to changes in their metabolic activity [70,72]. Understanding the effects of NNS on the gut microbiota, particularly the most commonly used like aspartame, acesulfame-k, sucralose, and saccharin, is critical for assessing their impact on human health. Due to the variety of NNS’ chemical structures and metabolism, their peculiar activities in the gut microbiome need to be better understood, too (Figure 3).” A new figure (Figure 3) has been added to show the chemical structures of NNS considered in this study.
In addition, the metabolism of the NNS considered has been described, on page 5, line 203, as follows: “Aspartame is structurally simple and consists of a dipeptide methyl ester containing two amino acids that are found widely in fruits, vegetables, nuts, and dairy products, namely, L-aspartic acid and L-phenylalanine (Figure 3) [85]. Upon consumption, gastrointestinal peptidases and esterases almost completely break down aspartame, resulting in negligible amounts of the compound entering the bloodstream [85,86]. An intestinal esterase, such as chymotrypsin, removes the methyl group and releases the natural dipeptide aspartyl phenylalanine. The microvillus mem-brane lining the small intestine then metabolizes this dipeptide into its component amino acids, which enter the circulatory system [87]. Phenylalanine in the liver can be converted to tyrosine by phenylalanine hydrolases. However, if the body does not require excess phenylalanine, it is eliminated in urine [88,89]. Once in systemic circulation, phenylalanine can be distributed throughout the body, including the brain, where it plays an important role in normal growth and development [85,88]. In the brain, tyrosine can further be converted into neurotransmitters such as dopamine, norepinephrine, and epinephrine. Thus, as such, aspartame in its whole form cannot interact directly with colonic microbiota. Acesulfame-K is a hydrophilic, organic acid derivative (Figure 3) absorbed almost entirely in the small intestine as an intact molecule and distributed by the blood to different tissues. Without undergoing any metabolism, more than 99% of ingested acesulfame- K is excreted in the urinary tract within the first 24 h, with less than 1% eliminated in the feces both in animals and humans [89,90]. Its rapid absorption and urinary excretion cause negligible acesulfame-K concentration that reaches the fecal or colonic bacteria [89,90]. Therefore, it is improbable that this NNS could directly affect the colonic microbiota [89,91].
Sucralose, a disaccharide composed of 1,6-dichloro-1,6-dideoxyfructose and 4-chloro-4-deoxygalactose (Figure 3), has a very low absorption level (less than 15%) and is practically not metabolized. Therefore, after intake, more than 85% of sucralose reaches the colon unchanged in all species, including humans [89,89,92]. The small proportion absorbed is then eliminated in the urine, mainly unchanged, though two glucuronides of sucralose were also detected in a small proportion (approximately 2%) [93]. Although more than 85% of the ingested sucralose contacts the colonic microbiota, 94%, and 99% are recovered in the feces without any structural change, thus indicating little or no intestinal metabolism [89,91].
Saccharin is an acid that dissolves in water (Figure 3), and it is more easily absorbed by animals with lower stomach pH such as humans and rabbits, compared to animals with higher stomach pH like rats [94,95]. In humans, between 85% and 95% of ingested saccharin is absorbed as an intact molecule since it does not undergo gastrointestinal metabolism. Once absorbed, it binds to plasma proteins, is distributed throughout the body, and is eliminated in urine through active tubular transport [89,94,96]. A small percentage of non-absorbed saccharin is excreted in the feces, indicating that high concentrations of this NNS could change the composition of the intestinal microbial population [91].”
In the Conclusions, line 705, the following paragraph has been added:” Different effects of NNS on the metabolism of the gut microbiota have been described till now. Although a negligible amount of ingested NNS can reach the intestine few studies here reported indicated that the gut microbiota can metabolize them producing a variety of biological effects summarized in Figure 5. NNS can be used as a carbon source by some strains of gut bacteria, leading to changes in their metabolic activity and modulating the production of SCFAs [70,72]. These compounds, such as acetate, propionate, and butyrate, have an impact on glucose metabolism, and host metabolism and exert an anti-inflammatory effect [130–132]. …… NNS can also alter the expression of genes involved in bacterial metabolism, altering the composition and function of the gut microbial community. They were also reported able to affect the release of gut hormones and neurotransmitters, influencing gut motility, nutrient absorption, and the composition of the gut microbiome thus inducing alterations in glucose metabolism. Some studies suggested that NNS can induce gut dysbiosis and inflammation by increasing the levels of bile acids [137,138].”
Q4. The authors should add a new figure to show the chemical structures of the non-nutritive sweeteners. This would help the readers to understand the sweeteners more easily.
A4. The chemical structures of NNS considered in this study have been reported in the new Figure 3.
Round 2
Reviewer 2 Report
The authors have critically revised the manuscript. I suggest to accept it.